# Clinical Usefulness of FRAX Score for Predicting Sarcopenia in Patients with Chronic Liver Disease

**DOI:** 10.3390/jcm10184080

**Published:** 2021-09-09

**Authors:** Chisato Saeki, Mitsuru Saito, Tomoya Kanai, Masanori Nakano, Tsunekazu Oikawa, Yuichi Torisu, Masayuki Saruta, Akihito Tsubota

**Affiliations:** 1Division of Gastroenterology and Hepatology, Department of Internal Medicine, The Jikei University School of Medicine, 3-25-8 Nishi-Shimbashi, Minato-ku, Tokyo 105-8461, Japan; tomoyaaust@hotmail.com (T.K.); masanori-nakano@jikei.ac.jp (M.N.); oitsune@jikei.ac.jp (T.O.); torisu@jikei.ac.jp (Y.T.); m.saruta@jikei.ac.jp (M.S.); 2Division of Gastroenterology, Department of Internal Medicine, Fuji City General Hospital, 50 Takashima-cho, Fuji-shi 417-8567, Shizuoka, Japan; 3Department of Orthopaedic Surgery, The Jikei University School of Medicine, 3-25-8 Nishi-Shimbashi, Minato-ku, Tokyo 105-8461, Japan; xlink67@gol.com; 4Core Research Facilities, Research Center for Medical Science, The Jikei University School of Medicine, 3-25-8 Nishi-Shimbashi, Minato-ku, Tokyo 105-8461, Japan

**Keywords:** chronic liver disease, osteoporotic fracture, Fracture Risk Assessment tool, sarcopenia

## Abstract

We investigated the usefulness of the Fracture Risk Assessment tool (FRAX) for predicting sarcopenia in chronic liver disease (CLD). In this cross-sectional study, we evaluated 321 patients with CLD. The FRAX with and without bone mineral density (BMD) was employed to calculate the 10-year risks of major osteoporotic and hip fractures. The FRAX score for high fracture risk was defined as a 10-year major osteoporotic fracture probability of ≥20% or a 10-year hip fracture probability of ≥3%. The diagnosis of sarcopenia was based on the Japan Society of Hepatology criteria. According to the FRAX, with and without BMD, 134 (41.7%) and 193 (60.1%) patients had a high fracture risk, respectively. The high fracture risk group had a significantly higher frequency of sarcopenia than the non-high fracture risk group. FRAX scores of major osteoporotic and hip fractures were negatively correlated with handgrip strength and muscle mass. Using the FRAX with BMD, the cutoff scores of major osteoporotic and hip fractures for predicting sarcopenia were 8.55% (sensitivity/specificity, 0.847/0.568) and 3.35% (0.729/0.746), respectively. Using the FRAX without BMD, they were 18.5% (0.635/0.725) and 7.65% (0.729/0.758), respectively. The FRAX is a simple and convenient screening tool for predicting sarcopenia in patients with CLD.

## 1. Introduction

The liver is a multifunctional organ involved in vitamin D metabolism, hormonal regulation, and cytokine production; therefore, its impaired function can dysregulate bone homeostasis [1]. Accordingly, osteoporosis is frequently noticed in patients with chronic liver disease (CLD), characterized by reduced bone mass and microarchitectural deterioration of bone, and resultant fragility fractures [1,2,3,4]. Bone disorders cause immobility and impaired physical function, thereby negatively affecting quality of life and prognosis [5,6].

The two major components of the musculoskeletal system are the bone and skeletal muscle, and their interaction, as mediated by mechanical and biochemical communication, is now being recognized [7,8,9,10]. Sarcopenia, characterized by decreased skeletal muscle mass and function, is closely associated with osteoporosis and fragility fractures in patients with CLD [11,12]. Consequently, the concept of osteosarcopenia has been established in order to represent this concomitant occurrence of sarcopenia and osteoporosis [13,14]. It is, therefore, crucial for patients with CLD to undergo comprehensive assessment of osteoporosis, fragility fractures, and sarcopenia.

In 2008, the World Health Organization (WHO) established the Fracture Risk Assessment tool (FRAX) to evaluate the 10-year probabilities of major osteoporotic and hip fractures [15]. The FRAX model comprises several risk factors for fragility fractures, such as sex, age, prevalent fractures, and the presence of diseases or conditions vulnerable to secondary osteoporosis. The National Osteoporosis Foundation recommends the initiation of pharmacologic osteoporosis treatment based on bone mineral density (BMD) values, as assessed using dual-energy X-ray absorptiometry (DEXA) and the FRAX algorithm: (i) T-score ≤ −2.5 (at the lumbar spine, femoral neck, or total hip); (ii) T-score between −1.0 and −2.5 in men aged ≥50 years and postmenopausal women; and (iii) a 10-year major osteoporotic or hip fracture probability ≥20% or ≥3%, respectively [16]. In one study of Indian patients with liver cirrhosis (LC) (with a median age of 49 years), the 10-year probabilities of hip and major osteoporotic fractures based on the FRAX algorithm with BMD were 2.5% and 5.7%, respectively [17]. In addition, approximately one-third of participants were at a high risk of fractures. In other studies of the general population, the FRAX score was independently associated with sarcopenia and could predict sarcopenia with high sensitivity [18,19]. Therefore, the FRAX algorithm might be useful as a simple and convenient screening tool for predicting sarcopenia, as well as estimating the probabilities of further fractures.

The present study aimed to evaluate the osteoporotic fracture risk using the FRAX algorithm and investigate whether the FRAX score could be useful for the prediction of sarcopenia in patients with CLD.

## 2. Materials and Methods

### 2.1. Participants and Study Design 

In this cross-sectional study, a total of 321 patients with CLD who visited Fuji City General Hospital (Shizuoka, Japan) and the Jikei University School of Medicine (Tokyo, Japan) between February 2017 and November 2020 were enrolled. The inclusion criteria were as follows: (1) men aged ≥50 years and postmenopausal women; (2) data from BMD measurement using DEXA (PRODIGY; GE Healthcare, Tokyo, Japan) were available; and (3) data from handgrip strength measurement using a hand dynamometer (T.K.K5401 GRIP-D; Takei Scientific Instruments, Niigata, Japan) and skeletal muscle mass index (SMI) measurement using bioelectrical impedance analysis (InBody S10; InBody, Seoul, Korea) were available. Patients with implants, hemodialysis, or massive ascites were excluded because of the unreliability of the bioelectrical impedance analysis method [11]. This study protocol was approved by the ethics committee of Fuji City General Hospital (approval no. 156) and the Jikei University School of Medicine (approval no. 28-196) and the study was conducted in accordance with the 2013 Declaration of Helsinki. 

### 2.2. Osteoporosis and Fracture Assessment 

BMD of the lumbar spine (L2–L4), femoral neck, and total hip was assessed using DEXA. The WHO criteria were adopted for the diagnosis of osteoporosis [20]. Information on the history of previous fractures was collected from medical records, medical interviews, and/or radiographs. Prevalent vertebral fractures were investigated semi-quantitatively using radiographs of the lateral thoracolumbar spine [21]. 

### 2.3. Sarcopenia and Gait Speed Assessment

The Japan Society of Hepatology criteria were adopted for the diagnosis of sarcopenia [22]. In brief, the reference values of reduced handgrip strength and muscle mass were <26 kg and SMI < 7.0 kg/m^2^ for men, and <18 kg and SMI < 5.7 kg/m^2^ for women, respectively. Gait speed was evaluated by walking 6 m, and slow gait speed was defined as less than 1.0 m/s.

### 2.4. Fracture Risk Assessment Based on the FRAX

The online FRAX with and without femoral neck BMD (Japan model: https://www.sheffield.ac.uk/FRAX/tool.aspx?country=3 (accessed on 1 July 2021)) was employed to calculate the 10-year probabilities of major osteoporotic and hip fractures (FRAX scores). FRAX scores take into account the following risk factors for fragility fracture: sex, age, weight, height, prevalent fractures, parental history of hip fracture, current excessive drinking (>3 units/day), current smoking status, glucocorticoid use, rheumatoid arthritis, presence of diseases or conditions related to secondary osteoporosis, such as CLD, and femoral neck BMD (if available) [15]. The FRAX score for high risk of osteoporotic fracture was defined as a 10-year major osteoporotic fracture probability ≥20% or a 10-year hip fracture probability ≥3% [16].

### 2.5. Biochemical Assessment

Serum total bilirubin, albumin, and prothrombin time-international normalized ratio (PT-INR) were measured using standard laboratory methods. In addition, serum branched chain amino acid (BCAA) and insulin like growth factor 1 (IGF-1) were measured using an enzymatic method (TOYOBO, Osaka, Japan) and an immunoradiometric assay (Fujirebio, Tokyo, Japan), respectively. BCAA and IGF-1 are involved in muscle protein synthesis via the activation of the mammalian target of the rapamycin pathway and are associated with sarcopenia [11,23].

### 2.6. Statistical Analysis

The Mann–Whitney U test was employed to assess the significance of differences in continuous variables. The chi-squared test was used to estimate the significance of differences in categorical variables. Spearman’s rank correlation test was carried out to evaluate the correlations between FRAX scores with and without BMD and continuous variables. For the prediction of sarcopenia, the receiver operating characteristic (ROC) curves of the FRAX score were employed to determine the optimal cutoff values using the Youden index [24]. All statistical analyses were carried out using SPSS version 26 software (IBM Japan, Tokyo, Japan). Where *p* values < 0.05, they were considered to be statistically significant.

## 3. Results

### 3.1. Patient Baseline Characteristics

Table 1 shows the baseline clinical characteristics of the 321 patients with CLD. The study cohort included 138 men (43.0%), with a median age of 70.0 (61.0–76.0) years. The number of patients diagnosed with LC was 158 (49.2%). The proportions of patients with sarcopenia, osteoporosis, and prevalent fractures were 26.5% (85/321), 32.7% (105/321), and 31.8% (102/321), respectively. The prevalence of sarcopenia was significantly higher in patients with LC than in those with non-LC (32.9% vs. 20.2%, *p* = 0.010; Appendix A), whereas the prevalence of osteoporosis was not significantly different between the two groups (33.5% vs. 31.9%, *p* = 0.754; Appendix A). The frequency of patients receiving pharmacological osteoporosis treatment was 28.7% (92/321). The 10-year probabilities of major osteoporotic and hip fractures in all subjects based on the FRAX algorithm with BMD were 9.3 (5.2–16.0) % and 1.9 (0.7–5.3) %, respectively (Appendix A). Meanwhile, using the FRAX algorithm without BMD, these probabilities were 14.0 (7.4–22.0) % and 4.6 (1.3–11.0) %, respectively (Appendix A). 

### 3.2. Comparison of Clinical Characteristics between Patients with and without High Fracture Risk Based on the FRAX with BMD

As shown in Table 1, when assessed using the FRAX with BMD, 134 (41.7%) patients were at a high risk of fractures. The 10-year probabilities of major osteoporotic and hip fractures in patients with a high fracture risk were 18.0 (13.0–27.3) % and 6.1 (4.4–11.3) %, respectively, while those not at high risk were 5.7 (4.1–8.1) % and 0.8 (0.3–1.7) %, respectively (Appendix A). Men accounted for 36.6% of the high fracture risk group and 47.6% of the non-high fracture risk group, with high fracture risk being less prevalent compared to women (*p* = 0.049). Between patients with and without a high fracture risk, significant differences were found in age (*p* < 0.001), body mass index (BMI; *p* = 0.001), IGF-1 (*p* < 0.001), and BCAA levels (*p* < 0.001). The BMD values were significantly lower in patients with a high fracture risk than in those without (*p* < 0.001 for all). SMI, handgrip strength, and gait speed values were also significantly lower in high-risk individuals (*p* < 0.001 for all). The frequencies of patients with osteoporosis (69.4% vs. 6.4%) and prevalent fracture (61.9% vs. 10.2%) were significantly higher in the high-risk group (*p* < 0.001 for both). Pharmacological osteoporosis treatment was received by 53.7% (72/134) of patients in the high-risk group and 10.7% (20/187) in the non-high-risk group (*p* < 0.001). Notably, patients with a high fracture risk showed a significantly higher prevalence of sarcopenia (47.0% vs. 11.8%), osteosarcopenia (36.6% vs. 2.7%), and slow gait speed (54.5% vs. 16.0%) than those without (*p* < 0.001 for all; Figure 1A–C).

### 3.3. Comparison of Clinical Characteristics between Patients with and without High Fracture Risk Based on the FRAX without BMD

As shown in Table 2, when assessed using the FRAX without BMD, 193 (60.1%) patients had a high fracture risk. The 10-year probabilities of major osteoporotic and hip fractures in patients with a high fracture risk were 21.0 (15.0–31.0) % and 9.5 (5.8–16.0) %, respectively, while those not at high risk were 6.8 (5.1–8.2) % and 1.1 (0.6–1.7) %, respectively (Appendix A). Between patients with and without a high fracture risk, significant differences were found in age (*p* < 0.001), BMI (*p* < 0.001), total bilirubin (*p* < 0.001), IGF-1 (*p* = 0.001), BCAA (*p* = 0.006), and PT-INR levels (*p* = 0.010). The high-risk group had higher glucocorticoid use (*p* = 0.005) and a lower alcohol intake (*p* = 0.010) than the non-high-risk group. A higher proportion of hepatitis B was found in the non-high-risk group. The frequencies of patients with osteoporosis (46.1% vs. 12.5%) and prevalent fracture (47.2% vs. 8.6%) were significantly higher in the high-risk group than in the non-high-risk group (*p* < 0.001 for both). The proportion of patients receiving pharmacological osteoporosis treatment was 40.9% (79/193) in patients with a high fracture risk and 10.2% (13/128) in those without (*p* < 0.001). It is worth noting that the prevalence of sarcopenia (37.3% vs. 10.2%), osteosarcopenia (24.9% vs. 4.7%), and slow gait speed (42.0% vs. 17.2%) was significantly higher in the high-risk group than in the non-high-risk group (*p* < 0.001 for all; Figure 2A–C).

### 3.4. Correlations between FRAX Score, SMI, and Handgrip Strength

The correlations between FRAX score, SMI, and handgrip strength were investigated using Spearman’s rank correlation test (Figure 3). In the FRAX algorithm with BMD, the FRAX scores of major osteoporotic and hip fractures were significantly correlated with SMI (*r* = −0.537 and −0.448, respectively, *p* < 0.001 for both; Figure 3A,B) and handgrip strength (*r* = −0.584 and −0.477, respectively, *p* < 0.001 for both; Figure 3C,D). Similarly, in the FRAX algorithm without BMD, the FRAX scores of major osteoporotic and hip fractures were significantly correlated with SMI (*r* = −0.556 and −0.524, respectively, *p* < 0.001 for both; Figure 3E,F) and handgrip strength (*r* = −0.564 and −0.496, respectively, *p* < 0.001 for both; Figure 3G,H).

### 3.5. Significant Factors Associated with Sarcopenia

On univariate analysis, the following nine factors were associated with sarcopenia: age, BMI, LC, albumin, IGF-1, BCAA, osteoporosis, prevalent fracture, and high fracture risk (Appendix A). On multivariate analysis, age (odds ratio [OR], 1.055; 95% confidence interval [CI], 1.014–1.097; *p* = 0.008), BMI (OR, 0.761; 95%CI, 0.681–0.852; *p* < 0.001), IGF-1 (OR, 0.980; 95%CI, 0.967–0.993; *p* = 0.003), BCAA (OR, 0.995; 95%CI, 0.991–0.999; *p* = 0.007), and high fracture risk based on the FRAX (OR, 3.143; 95%CI, 1.559–6.340; *p* = 0.001) remained significant and independent factors associated with sarcopenia (Appendix A).

### 3.6. Optimal Cutoff of FRAX Score for Predicting Sarcopenia

For the prediction of sarcopenia, an ROC curve analysis was carried out to determine the optimal cutoff values of FRAX score for major osteoporotic and hip fractures (Figure 4). In the FRAX with BMD, the area under curve (AUC) values of major osteoporotic and hip fractures were 0.74 and 0.78, respectively (Figure 4A,B). The cutoff values for predicting sarcopenia were 8.55% and 3.35%, respectively, with the sensitivity, specificity, positive predictive value (PPV), and negative predictive value (NPV) being 0.847 and 0.729, 0.568 and 0.746, 0.414 and 0.508, and 0.912 and 0.884, respectively. Similarly, in the FRAX without BMD, the AUC values of major osteoporotic and hip fractures were 0.74 and 0.78, respectively (Figure 4C,D). The cutoff values were 18.5% and 7.65%, respectively, with the sensitivity, specificity, PPV, and NPV being 0.635 and 0.729, 0.725 and 0.758, 0.473 and 0.521, and 0.844 and 0.886, respectively.

## 4. Discussion

Owing to the close relation of the bone and muscle during development and growth, osteoporosis, osteoporotic fracture, sarcopenia, and osteosarcopenia, which are included in musculoskeletal disorders, often coexist and progress in parallel [8,9,10,11]. These disorders, which are strongly associated with each other, are frequently noticed in patients with CLD [12,13]. Therefore, early comprehensive assessment, therapeutic intervention, and prevention of these disorders are essential, especially in patients with CLD. In this study, we investigated the 10-year probabilities of major osteoporotic and hip fractures based on the FRAX algorithm and the usefulness of the FRAX score for the prediction of sarcopenia in patients with CLD.

The FRAX score has been reported to predict the incidence of sarcopenia in the general population [19,20]. Our study showed that patients at a high risk of fracture had a significantly higher frequency of sarcopenia, osteosarcopenia, and a slow gait speed than those without the risk. The FRAX scores were significantly and inversely associated with SMI and handgrip strength. Multivariate analysis identified high fracture risk based on the FRAX as an independent factor associated with sarcopenia. The ROC curve analysis showed the cutoff FRAX scores of major osteoporotic and hip fractures for predicting sarcopenia to be 8.55% (sensitivity/specificity, 0.847/0.568) and 3.35% (0.729/0.746), respectively, when calculated with BMD; these values were 18.5% (0.635/0.725) and 7.65% (0.729/0.758), respectively, when calculated without BMD. These results suggest that the FRAX algorithm could be helpful for predicting sarcopenia in patients with CLD.

The FRAX algorithm includes items related to both osteoporotic fracture and sarcopenia, such as age, weight and height (i.e., BMI), prevalent fractures, smoking habit, and alcohol intake [15,18]. A previous report revealed that patients with prevalent vertebral fractures are at an increased risk of developing further fractures, even after adjusting for confounding factors [25]. Importantly, vertebral and hip fractures could cause impairment in physical function and immobility, thereby leading to sarcopenia [5,26,27]. Therefore, prevalent fractures are a common risk factor for the incidence of further fractures and sarcopenia.

The European Working Group on Sarcopenia in Older People recommends the use of the self-reported SARC-F questionnaire as a simple screening tool for sarcopenia, which comprises the following items: strength (S), assistance in walking (A), rising from a chair (R), climbing stairs (C), and fall history (F) [28,29]. A previous study demonstrated that the SARC-F had excellent specificity (94–99%), but extremely low sensitivity (3.8–9.9%) against three consensus definitions of sarcopenia from international, Asian, and European working groups on sarcopenia [30]. A meta-analysis of seven studies, including a total of 12,800 subjects, demonstrated that the SARC-F had high specificity (90%) and low sensitivity (21%) [31]. In contrast to the SARC-F, our study showed that the FRAX algorithm with and without BMD could predict sarcopenia with lower specificity (57–76%) and higher sensitivity (64–85%). As the FRAX score without BMD is easily computed with an online calculator and without specialized equipment, the FRAX is a simple and convenient screening tool for sarcopenia. Given that the prevalence of sarcopenia is high especially in patients with LC (as also shown in this study) [22], the FRAX is useful for predicting sarcopenia in such patients.

There were some limitations in this cross-sectional study. First, we did not prospectively assess the incidence of the osteoporotic fractures and sarcopenia over a long-term observation period. A large-scale prospective study is required to confirm the usefulness of the FRAX algorithm for predicting fractures and sarcopenia in CLD. Second, the nutritional intake and daily physical activity of patients, which may affect the development of sarcopenia, were not taken into account. Lastly, this study did not include healthy subjects as controls.

## 5. Conclusions

In this study, we demonstrated that FRAX score could be helpful for predicting sarcopenia in patients with CLD. The FRAX algorithm is a simple and convenient screening method for predicting sarcopenia in clinical settings without specialized equipment.

## Figures and Tables

**Figure 1 jcm-10-04080-f001:**
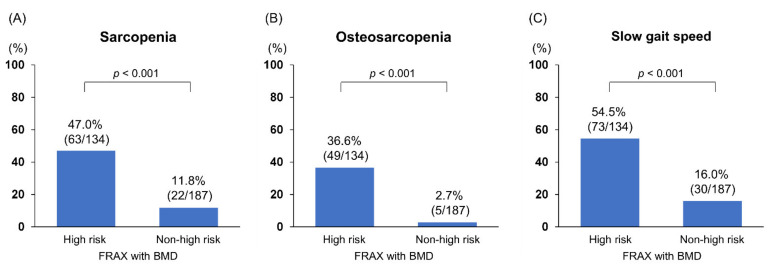
The frequency of sarcopenia, osteosarcopenia, and slow gait speed between patients with and without high fracture risk based on the Fracture Risk Assessment tool with bone mineral density. The frequency of (**A**) sarcopenia, (**B**) osteosarcopenia, and (**C**) slow gait speed was significantly higher in the high fracture risk group than in the non-high fracture risk group (*p* < 0.001 for all).

**Figure 2 jcm-10-04080-f002:**
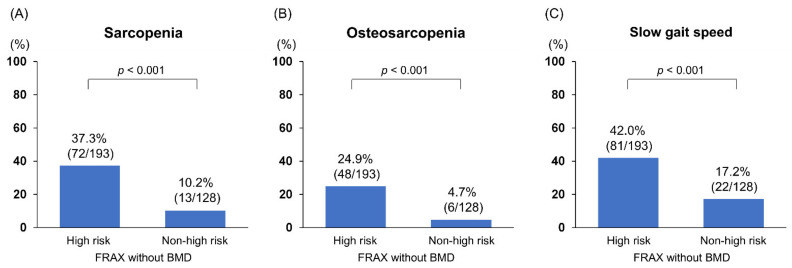
The frequency of sarcopenia, osteosarcopenia, and slow gait speed between patients with and without high fracture risk based on the Fracture Risk Assessment tool without bone mineral density. The frequency of (**A**) sarcopenia, (**B**) osteosarcopenia, and (**C**) slow gait speed was significantly higher in the high fracture risk group than in the non-high fracture risk group (*p* < 0.001 for all).

**Figure 3 jcm-10-04080-f003:**
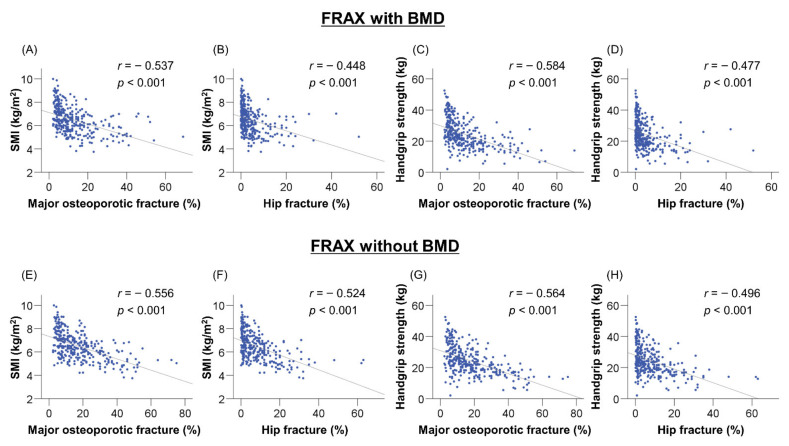
Correlations between the Fracture Risk Assessment tool (FRAX) scores of major osteoporotic fracture and hip fracture with and without bone mineral density (BMD) and skeletal muscle mass index (SMI) and handgrip strength. The FRAX scores of major osteoporotic and hip fractures with BMD were significantly correlated with the (**A**,**B**) SMI and (**C**,**D**) handgrip strength (*p* < 0.001 for all). Similarly, the FRAX scores of major osteoporotic and hip fractures without BMD were significantly correlated with the (**E**,**F**) SMI and (**G**,**H**) handgrip strength (*p* < 0.001 for all).

**Figure 4 jcm-10-04080-f004:**
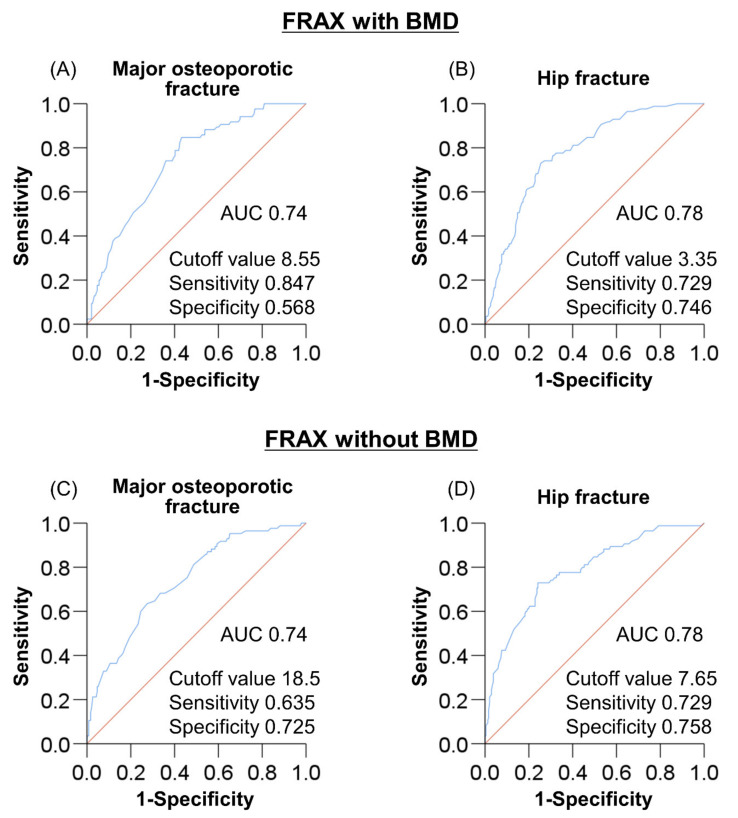
The receiver operating characteristic (ROC) curve analyses of Fracture Risk Assessment tool (FRAX) scores for predicting sarcopenia. (**A**,**B**) Based on the FRAX with bone mineral density (BMD), the cutoff values of FRAX score for major osteoporotic and hip fractures were 8.55% and 3.35%, respectively, with an area under the curve (AUC), sensitivity, and specificity of 0.74 and 0.78, 0.847 and 0.729, and 0.568 and 0.746, respectively. (**C**,**D**) Based on the FRAX without BMD, the cutoff values of FRAX score for major osteoporotic and hip fractures were 18.5% and 7.65%, respectively, with an AUC, sensitivity, and specificity of 0.74 and 0.78, 0.635 and 0.729, and 0.725 and 0.758, respectively.

**Table 1 jcm-10-04080-t001:** Comparison of clinical characteristics between patients with and without high fracture risk based on the FRAX with BMD.

Variable	All Patients	FRAX with BMD High Risk	FRAX with BMD Non-High Risk	*p* Value
Patients, *n* (%)	321	134 (41.7)	187 (58.3)	
Men, *n* (%)	138 (43.0)	49 (36.6)	89 (47.6)	0.049
Age (years)	70.0 (61.0–76.0)	75.0 (71.0–80.0)	65.0 (58.0–71.0)	<0.001
BMI (kg/m^2^)	23.0 (20.7–25.9)	22.2 (20.2–24.7)	23.6 (21.4–26.2)	0.001
Liver cirrhosis, *n* (%)	158 (49.2)	70 (52.2)	88 (47.1)	0.360
Glucocorticoid use, *n* (%)	20 (6.2)	11 (8.2)	9 (4.8)	0.214
Smoking, *n* (%)	78 (24.3)	31 (23.1)	47 (25.1)	0.680
Alcohol intake, *n* (%)	33 (10.3)	11 (8.2)	22 (11.8)	0.887
Etiology				
HBV/HCV/alcohol/other, *n*	37/96/52/136	12/49/16/57	25/47/36/79	0.063
Total bilirubin (mg/dL)	0.7 (0.5–1.0)	0.7 (0.5–0.9)	0.7 (0.6–1.1)	0.020
Albumin (g/dL)	4.0 (3.7–4.3)	4.0 (3.8–4.3)	4.1 (3.7–4.3)	0.900
Prothrombin time (%)	92 (77–100)	93 (81–100)	92 (75–100)	0.339
IGF-1 (ng/mL)	66 (48–88)	60 (46–77)	74 (51–96)	<0.001
BCAA (µmol/L)	408 (352–470)	381 (326–435)	435 (385–489)	<0.001
Lumbar spine BMD (g/cm^2^)	1.07 (0.89–1.22)	0.92 (0.81–1.08)	1.15 (1.01–1.26)	<0.001
Femoral neck BMD (g/cm^2^)	0.75 (0.67–0.88)	0.65 (0.59–0.70)	0.85 (0.77–0.94)	<0.001
Total hip BMD (g/cm^2^)	0.82 (0.71–0.93)	0.70 (0.64–0.77)	0.90 (0.82–0.98)	<0.001
SMI (kg/m^2^)				
All patients	6.43 (5.67–7.18)	5.89 (5.09–6.77)	6.72 (6.03–7.63)	<0.001
Men	7.18 (6.58–7.80)	6.75 (6.12–7.26)	7.43 (6.99–8.21)	<0.001
Women	5.95 (5.27–6.49)	5.39 (4.97–6.09)	6.12 (5.72–6.57)	<0.001
Handgrip strength (kg)				
All patients	22.7 (18.1–30.1)	18.5 (14.9–23.5)	25.7 (21.7–33.2)	<0.001
Men	30.9 (24.4–37.4)	25.4 (20.7–31.9)	34.2 (27.6–39.8)	<0.001
Women	19.5 (16.5–22.6)	17.0 (14.0–19.4)	22.3 (19.1–24.9)	<0.001
Gait speed (m/s)	1.11 (0.93–1.28)	0.99 (0.77–1.14)	1.20 (1.03–1.38)	<0.001
Sarcopenia, *n* (%)	85 (26.5)	63 (47.0)	22 (11.8)	<0.001
Osteoporosis, *n* (%)	105 (32.7)	93 (69.4)	12 (6.4)	<0.001
Prevalent fracture, *n* (%)	102 (31.8)	83 (61.9)	19 (10.2)	<0.001

Values are shown as median (interquartile range) or number (percentage). Statistical analysis was performed using the chi-squared test or the Mann–Whitney U test, as appropriate. FRAX, Fracture Risk Assessment tool; BCAA, branched-chain amino acid; BMD, bone mineral density; BMI, body mass index; HBV, hepatitis B virus; HCV, hepatitis C virus; IGF-1, insulin-like growth factor 1; SMI, skeletal muscle mass index.

**Table 2 jcm-10-04080-t002:** Comparison of clinical characteristics between patients with and without high fracture risk based on the FRAX without BMD.

Variable	FRAX without BMD High Risk	FRAX without BMD Non-High Risk	*p* Value
Patients, *n* (%)	193 (60.1)	128 (39.9)	
Men, *n* (%)	76 (39.4)	62 (48.4)	0.108
Age (years)	75.0 (71.0–79.0)	59.0 (55.2–65.0)	<0.001
BMI (kg/m^2^)	22.3 (20.2–24.4)	24.5 (22.1–27.0)	<0.001
Liver cirrhosis, *n* (%)	95 (49.2)	63 (49.2)	0.999
Glucocorticoid use, *n* (%)	18 (9.3)	2 (1.6)	0.005
Smoking, *n* (%)	42 (21.8)	36 (28.1)	0.193
Alcohol intake, *n* (%)	13 (6.7)	20 (15.6)	0.010
Etiology			
HBV/HCV/alcohol/other, *n*	15/65/25/88	22/31/27/48	0.006
Total bilirubin (mg/dL)	0.7 (0.5–0.9)	0.8 (0.6–1.3)	<0.001
Albumin (g/dL)	4.0 (3.8–4.3)	4.1 (3.6–4.4)	0.220
Prothrombin time (%)	95 (81–100)	87 (74–100)	0.010
IGF-1 (ng/mL)	61 (47–83)	75 (52–101)	0.001
BCAA (µmol/L)	394 (342–456)	436 (382–490)	0.006
Lumbar spine BMD (g/cm^2^)	1.00 (0.85–1.18)	1.11 (0.97–1.26)	<0.001
Femoral neck BMD (g/cm^2^)	0.70 (0.63–0.82)	0.84 (0.74–0.93)	<0.001
Total hip BMD (g/cm^2^)	0.76 (0.69–0.85)	0.89 (0.81–0.99)	<0.001
SMI (kg/m^2^)			
All patients	6.09 (5.32–6.82)	7.07 (6.30–7.86)	<0.001
Men	6.90 (6.20–7.36)	7.79 (7.13–8.50)	<0.001
Women	5.74 (5.09–6.23)	6.32 (5.75–6.60)	<0.001
Handgrip strength (kg)			
All patients	20.7 (17.0–25.8)	27.0 (21.9–36.6)	<0.001
Men	27.0 (22.8–32.0)	36.8 (30.5–40.7)	<0.001
Women	18.3 (15.0–21.7)	22.5 (18.4–25.8)	<0.001
Gait speed (m/s)	1.06 (0.86–1.22)	1.18 (1.03–1.36)	<0.001
Sarcopenia, *n* (%)	72 (37.3)	13 (10.2)	<0.001
Osteoporosis, *n* (%)	89 (46.1)	16 (12.5)	<0.001
Prevalent fracture, *n* (%)	91 (47.2)	11 (8.6)	<0.001

Values are shown as median (interquartile range) or number (percentage). Statistical analysis was performed using the chi-squared test or the Mann–Whitney U test, as appropriate. FRAX, Fracture Risk Assessment tool; BCAA, branched-chain amino acid; BMD, bone mineral density; BMI, body mass index; HBV, hepatitis B virus; HCV, hepatitis C virus; IGF-1, insulin-like growth factor 1; SMI, skeletal muscle mass index.

## Data Availability

The data presented in this study are available on request from the corresponding author. The data are not publicly available due to patients’ privacy.

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
