# Peer review of "Clinical Usefulness of FRAX Score for Predicting Sarcopenia in Patients with Chronic Liver Disease"

_jcm, 2021, doi:10.3390/jcm10184080_

Round 1

Reviewer 1 Report

 I have read this paper with great interest, and I think it is meaningful because it presents applicable results in actual clinical settings. 

  1. Can the authors further discuss the characteristics of people with sarcopenia in a non-high risk group? I am also concerned that the issue may be blurred and the discussion may be prolonged. If additional discussion is likely to get in the way, I would like you to deal with that in other next paper.
  2. The sarcopenia definition differs from the cut-values suggested by the ASIA working group(2019). It would be nice if you could add a brief comment on this.

Author Response

RESPONSES TO THE REVIEWER

We wish to express our deep appreciation to the reviewers for the constructive comments on our manuscript. Our point-by-point responses to the reviewers’ comments are listed below.

Comments and Suggestions for Authors

Reviewer #1: I have read this paper with great interest, and I think it is meaningful because it presents applicable results in actual clinical settings. 

  1. Can the authors further discuss the characteristics of people with sarcopenia in a non-high risk group? I am also concerned that the issue may be blurred and the discussion may be prolonged. If additional discussion is likely to get in the way, I would like you to deal with that in other next paper.

Responses: We deeply appreciate the reviewer’s constructive suggestions and keen insight. We performed additional analyses to compare the clinical characteristics between patients with and without sarcopenia, who are not at a high fracture risk based on the FRAX with and without BMD (Tables R1 and R2 [only for review], respectively). Using the FRAX with BMD, patients with sarcopenia were older and had lower BMI than those without sarcopenia. Regarding biochemical parameters, the sarcopenia group had significantly lower levels of albumin, IGF-1, and BCAA than the non-sarcopenia group (Table R1). While, using the FRAX without BMD, patients with sarcopenia had significantly lower BMI, levels of albumin, IGF-1, and BCAA, and BMD (Table R2). Whichever scoring system (with or without MBD) was used, the sarcopenia group showed a significantly higher prevalence of osteoporosis than the non-sarcopenia group (Tables R1 and R2). As concerned by the reviewer, we afraid that adding the new results to the Results section would be complicated and might obscure the gist of this manuscript. Accordingly, we did not dare to add the new results to the revised text. In the future, we will address and discuss these issues in another study. We thank you again for the reviewer’s critical comments.

  1. The sarcopenia definition differs from the cut-values suggested by the ASIA working group (2019). It would be nice if you could add a brief comment on this.

Responses: Thank you for providing an interesting topic. As you may have known, the reference values of the Asian Working Group for Sarcopenia (AWGS) 2019 criteria for reduced muscle mass (as assessed by BIA method) are identical to those of the Japan Society of Hepatology (JSH) criteria (SMI <7.0 kg/m2 for men and <5.7 kg/m2 for women). Additionally, the reference value of reduced handgrip strength for women established by the AWGS 2019 criteria are also identical to that of JSH criteria (18 kg for women). However, the two criteria for men are slightly different (28 kg in AWGS 2019 and 26 kg in JSH). In a recent study of Japanese patients with CLD, low muscle strength was an independent predictor of mortality (Nishikawa H, et al. Hepatol Res 2021 ahead of print). The cutoff values of handgrip strength for prognosis were 27.8kg for men and 18.8kg for women, which are similar to the reference values of the AWGS 2019 criteria. Therefore, the JSH working group on sarcopenia has advocated the revised criteria of the handgrip strength for sarcopenia in patients with chronic liver disease (28 kg for men and 18 kg for women). We agree with the revision of the JSH criteria from a prognostic perspective.

Postscript: We have revised the abstract without changing the content.

Reviewer 2 Report

It is a well presented study about the connection between osteoporosis and sarcopenia in patients with chronic liver disease

I have some comments to make

  1. A comparison between patients with liver disease and a control group regarding osteoporosis (dexa, frax score, BMD) and sarcopenia is missing.
  2. In your cohort of patients a big proportion of them are patients with liver cirrhosis. You do not analyze if patients with liver cirrhosis have milder or more serious problems concerning osteoporosis and sarcopenia compared to patients without liver cirrhosis. Additionally you must study if the stage of liver cirrhosis is connected with the degree of osteoporosis and sarcopenia
  3. You calculated BCAA and IGF 1 but you did not explain why. Furthermore you did not explain if your results regarding these 2 molecules had any importance or any connection with osteoporosis and sarcopenia.
  4. You must study which are the independent risk factors related with sarcopenia. A multivariate analysis could be important 
  5. As the tests used for the diagnosis of sarcopenia are subjective enough , i believe that more objective parameters like measurement of iliopsoas muscle by using a CT scan are needed

Author Response

RESPONSES TO THE REVIEWER

We wish to express our deep appreciation to the reviewers for the constructive comments on our manuscript. Our point-by-point responses to the reviewers’ comments are listed below.

Comments and Suggestions for Authors

Reviewer #2: It is a well presented study about the connection between osteoporosis and sarcopenia in patients with chronic liver disease

I have some comments to make

1. A comparison between patients with liver disease and a control group regarding osteoporosis (dexa, frax score, BMD) and sarcopenia is missing.

Responses: We appreciate the reviewer’s critical comments. As pointed out by the reviewer, the current study did not include a healthy control group. Referring to the previous reports and meta-analysis, patients with liver disease are more susceptible to sarcopenia and osteoporosis than healthy individuals. However, there are no reports that directly compare the frequency of these complications or FRAX scores between patients with liver disease and healthy subjects. In the future, we are willing to conduct studies to directly assess the frequency of sarcopenia and bone diseases (including FRAX scores) among patients with liver disease and healthy controls. We newly added the limitation that this study did not include healthy subjects as controls to the Discussion section.

2. In your cohort of patients a big proportion of them are patients with liver      cirrhosis. You do not analyze if patients with liver cirrhosis have milder or more serious problems concerning osteoporosis and sarcopenia compared to patients without liver cirrhosis. Additionally you must study if the stage of liver cirrhosis is connected with the degree of osteoporosis and sarcopenia

Responses: We are thankful for the reviewer’s critical suggestions. We performed an additional analysis to compare the prevalence of sarcopenia and osteoporosis between patients with and without LC (Figure R1 [only for review]). In addition, we investigated the frequency of these complications between patients with compensated LC (Child-Pugh class A) and decompensated LC (Child-Pugh class B/C). The prevalence of sarcopenia was significantly higher in patients with LC than in those with non-LC (Figure R1 A), whereas the prevalence of osteoporosis was not significantly different between the two groups (Figure R1 B). The prevalence of these complications was not significantly different between patients with compensated and decompensated LC (Figure R1 C, D). We think that adding the new results to the Results section might obscure the gist of this manuscript. Accordingly, we did not add the new results to the revised text.

3. You calculated BCAA and IGF 1 but you did not explain why. Furthermore you did not explain if your results regarding these 2 molecules had any importance or any connection with osteoporosis and sarcopenia.

Responses: We thank the reviewer for insightful comments. BCAA and IGF-1 increase muscle protein synthesis via the activation of the mTOR pathway. Indeed, multivariate analysis identified lower BCAA and IGF-1 levels as independent factors associated with sarcopenia (new Table S4). However, we have previously and repeatedly reported the association between BCAAs, IGF-1, sarcopenia, and osteosarcopenia in patients with CLD (Saeki C, et al. BMC Musculoskelet Disord. 2019;26;615; Saeki C, et al. J Clin Med. 2020;9:2381; Saeki C, et al. J Clin Med. 2020;9:3239). In addition, the main purpose of this study was to investigate the relationship between FRAX score and sarcopenia, and to evaluate whether the FRAX score could be useful for predicting sarcopenia in patients with CLD. Therefore, we have not thoroughly discussed in depth the importance of BCAA and IGF-1 in the development of sarcopenia in this study.

4. You must study which are the independent risk factors related with sarcopenia. A multivariate analysis could be important.

Responses: As suggested by the reviewer, we performed univariate and multivariate analyses to identify independent factors associated with sarcopenia (new Table S3, S4). As a result, advanced age, lower BMI, IGF-1 levels, and BCAA levels, and high fracture risk based on the FRAX (with BMD) were independent factors associated with sarcopenia. These results suggest that high fracture risk based on FRAX could be an important factor for the development of sarcopenia. We newly added these results to the Results section (lines 212–220 in revised manuscript).

5. As the tests used for the diagnosis of sarcopenia are subjective enough, I believe that more objective parameters like measurement of iliopsoas muscle by using a CT scan are needed

Responses: We agree with the reviewer’s comments. Patients with CLD frequently undergo CT scans for surveillance of hepatocellular carcinoma. Therefore, the JSH criteria adopts the CT method to assess muscle mass in the clinical setting. Certainly, muscle mass measured by CT method is reproducible, accurate, and objective. However, specific software is needed to measure the cross-sectional area of skeletal muscle at the level of L3. Therefore, we could not assess muscle mass using the CT method. The SMI calculated by the BIA method is strongly correlated with the SMI calculated by the CT method (Nishikawa H, et al. Hepatol. Res. 2016;46:951–63). Furthermore, patients with implants, hemodialysis, or massive ascites were excluded from this study, because of the unreliability of the BIA method. Therefore, the muscle mass assessed by the BIA method in this study is reliable.

Postscript: We have revised the abstract without changing the content.

Round 2

Reviewer 2 Report

I am not fully satisfied with the answers given from the authors

I believe that as you decided to estimate parameters like glp 1 and bcaa  you must explain in your manuscript why you did that

The diferences between patients with and without liver cirrhosis must be added. I insist on that. Moreover, you must add 2-3 words about these results in the discussion

I agree with your decision to not present the comparison between patients with liver cirrhosis child A and those with liver cirrhosis child b/c as the number of patients is small and for this reason the results are questionable

Author Response

RESPONSES TO THE REVIEWER

We wish to express our deep appreciation to the reviewers for the constructive comments on our manuscript. Our point-by-point responses to the reviewers’ comments are listed below.

Comments and Suggestions for Authors

Reviewer #2: I am not fully satisfied with the answers given from the authors

  1. I believe that as you decided to estimate parameters like glp 1 and bcaa you must explain in your manuscript why you did that.

Responses: We appreciate the reviewer’s critical comments. BCAA and IGF-1 were measured because they are involved in muscle protein synthesis via the activation of the mTOR pathway. We newly added this description to the Methods section (lines 111–116 in revised manuscript).

  1. The diferences between patients with and without liver cirrhosis must be added. I insist on that. Moreover, you must add 2-3 words about these results in the discussion.

Responses: We are thankful for the reviewer’s critical suggestions. The prevalence of sarcopenia was significantly higher in patients with LC than in those with non-LC (new Figure S1A), whereas the prevalence of osteoporosis was not significantly different between the two groups (new Figure S1B). We newly added these results to the Results section (lines 132–135 in revised manuscript). Based on these results, we consider that FRAX is useful for predicting sarcopenia especially in patients with LC. We newly added this description to the Discussion section (lines 290–292 in revised manuscript).

  1. I agree with your decision to not present the comparison between patients with liver cirrhosis child A and those with liver cirrhosis child b/c as the number of patients is small and for this reason the results are questionable

Responses: We are thankful for agreeing to our proposal.

This manuscript is a resubmission of an earlier submission. The following is a list of the peer review reports and author responses from that submission.